# Design of Experiments to Achieve an Efficient Chitosan-Based DNA Vaccine Delivery System

**DOI:** 10.3390/pharmaceutics13091369

**Published:** 2021-08-31

**Authors:** Carlos Rodolfo, Dalinda Eusébio, Cathy Ventura, Renato Nunes, Helena F. Florindo, Diana Costa, Ângela Sousa

**Affiliations:** 1CICS-UBI—Health Science Research Centre, University of Beira Interior, Av. Infante D. Henrique, 6200-506 Covilhã, Portugal; carlos.rodolfo@ubi.pt (C.R.); dalinda-21@hotmail.com (D.E.); cathy.ventura@ubi.pt (C.V.); renatopereiranunes@hotmail.com (R.N.); dcosta@fcsaude.ubi.pt (D.C.); 2Research Institute for Medicines (iMed.ULisboa), Faculty of Pharmacy, Universidade de Lisboa, 1649-003 Lisbon, Portugal; hflorindo@ff.ulisboa.pt

**Keywords:** chitosan polymers, delivery systems, design of experiments, DNA vaccine, HPV

## Abstract

In current times, DNA vaccines are seen as a promising approach to treat and prevent diseases, such as virus infections and cancer. Aiming at the production of a functional and effective plasmid DNA (pDNA) delivery system, four chitosan polymers, differing in the molecular weight, were studied using the design of experiments (DoE) tool. These gene delivery systems were formulated by ionotropic gelation and exploring the chitosan and TPP concentrations as DoE inputs to maximize the nanoparticle positive charge and minimize their size and polydispersity index (PDI) as DoE outputs. The obtained linear and quadratic models were statistically significant (*p*-value < 0.05) and non-significant lack of fit, with suitable coefficient of determination and the respective optimal points successfully validated. Furthermore, morphology, stability and cytotoxicity assays were performed to evaluate the endurance of these systems over time and their further potential for future in vitro studies. The subsequent optimization process was successful achieved for the delivery systems based on the four chitosan polymers, in which the smallest particle size was obtained for the carrier containing the 5 kDa chitosan (~82 nm), while the nanosystem prepared with the high molecular weight (HMW) chitosan displayed the highest zeta potential (~+26.8 mV). Delivery systems were stable in the formulation buffer after a month and did not exhibit toxicity for the cells. In this sense, DoE revealed to be a powerful tool to explore and tailor the characteristics of chitosan/pDNA nanosystems significantly contributing to unraveling an optimum carrier for advancing the DNA vaccines delivery field.

## 1. Introduction

Human papillomavirus (HPV) is responsible for triggering a vast roll of cancers, among them the cervical, vaginal, vulval, penile, oropharynx and anal cancers. HPV has more than 150 genotypes completely sequenced and are divided into high and low-risk groups, according to their oncogenic potential. In the high-risk group, the subtypes HPV-16 and HPV-18 are responsible for 70% of all cervical cancers worldwide. For this reason, intensive studies have been devised to identify the pathways whereby carcinogenesis and infections are induced by these two HPV subtypes [1,2,3]. The E6 and E7 proteins were recognized as the major contributors for the development of cervical cancer in HPV-infected cells. These oncoproteins have the ability to deactivate the p53 and pRb tumor suppressor pathways, leading to uncontrolled cellular proliferation, invasion, metastasis, angiogenesis and unrestricted telomerase activity [4,5].

Three prophylactic vaccines are currently available to prevent HPV infection, Cervarix, Gardasil and Gardasil-9. The purpose of these vaccines is to prevent the virus propagation by inducing antibody responses against the L1 protein virus-like particle, which is present in the HPV viral capsid [6,7]. However, these vaccines only induce humoral immunity, producing neutralizing antibodies to prevent future infections, and for that reason, they are ineffective at facing pre-established infections. As a result, there is an urgent need for the development of therapeutic vaccines to treat cervical cancer. DNA vaccines can be a potential solution, as they can induce both humoral and cellular immune responses, resulting in preventive and therapeutic effects. These vaccines will present the encoded antigens to CD4+ T lymphocytes, which activate B cells to produce neutralizing antibodies, and to CD8+ T lymphocytes, which activate cytotoxic T cells and the effector cell maturation [8]. Furthermore, DNA vaccines are safe (by only carrying specific antigen genetic information), stable, in addition of being easy to prepare and produce at large scale. Given that E6 and E7 tumor antigens are constitutively expressed on the surface of HPV-infected cervical cancer cells, these are ideal antigen candidates to be encoded on DNA vaccines [2]. However, the transfection efficiency of eukaryotic cells with naked nucleic acids, such as plasmid DNA, is disfavored by the negative repulsion of cell membrane. In this field, several biomaterials based on cationic liposomes [9,10], peptides [11,12] and polymers [13,14] have been shown to be able to efficiently condense, protect, carry, and deliver nucleic acids to eukaryotic cells. For instance, chitosan and chitosan derivatives have been used as DNA delivery systems due to their cationic charge, biocompatibility, biodegradability, mucoadhesiveness and permeability-enhancing properties [15,16,17,18,19]. Mucoadhesive particles will integrate in the mucus layer via interactions with mucin fibers or electrostatic interactions with the negatively charged mucous layer to improve the entrance of molecules through the mucosal surface. They prolong the residence time in the mucosal areas, favoring a more effective absorption and controlled release of loaded pharmaceutics [18,20]. In this context, chitosan polymer is the suitable material to develop delivery nanosystems for DNA vaccines, when the intention is a needle free administration by mucosal direct application. This vaccination modality can be useful for vaginal mucosa administration of vaccines against HPV or intranasal mucosa administration of vaccines against respiratory virus (such as SARS-CoV-2), for instance, since these sites are the main entry routes of these viruses. Furthermore, this mucosal vaccination approach will be a safer and efficient strategy to elicit both systemic and mucosal immunity, comparing to parenteral administration, reducing pain, stresses and biohazards associated with needle-based injection [18,21]. Sodium tripolyphosphate (TPP) is an anion that has been used in conjugation with delivery systems. These two biomaterials can develop suitable DNA delivery systems by using the ionotropic gelation technique. With this procedure, the positively charged chitosan links through ionic cross-linking with the negatively charged TPP, forming nanoparticles [22,23,24,25]. Based on these considerations, the conjugation of chitosan with TPP can be a promising strategy for the formulation of DNA vaccines delivery systems.

Design of experiments (DoE) is an efficient analysis tool that can be used to optimize nanoparticles formulation [26,27], in a faster and easier manner, when compared with the common random experiment approach [28]. This tool allows the systematic and simultaneous variation of several parameters, applying a statistical model to use few experiments as possible [29,30]. This approach also considers the interaction effects between parameters, which sometimes is essential for better optimizing a process. Several models are available to apply in DoE and their selection depends on the aim of the work and the number and level of inputs. Thus, the Central Composite Face (CCF) model is composed of a full or fractional factorial design and start points placed on the faces of the sides. This model is useful to perform optimizations and can predict high-quality results, since it only uses points within the predetermined range [31].

Therefore, the main purpose of this work was to optimize the formulation of delivery systems based on chitosan polymers with different molecular weights and conjugated with TPP, exploring the CCF model, to deliver a pDNA vaccine encoding HPV E7 oncoprotein. This DoE model predicted the ideal conjugation of inputs (chitosan and TPP concentration) to optimize the defined outputs/responses (the lowest sizes, the highest zeta potentials, and a suitable polydispersity index (PDI)) to attain ideal chitosan/TPP/pDNA delivery systems. Various assays defined by DoE were explored and the resultant formulations were characterized in terms of their physicochemical characteristics (size, potential zeta and PDI) to introduce the respective responses into DoE. After the prediction and validation of the optimal point to each polymer, the morphology and stability (through the time at different temperatures and incubated with different culture mediums) of the optimized formulations were studied. Furthermore, transfection studies were performed to assess the nanoparticles’ biocompatibility.

## 2. Materials and Methods

### 2.1. Materials

Medical grade high molecular weight (HMW) chitosan, with a molecular weight range between 200 and 500 kDa, was purchased from Heppe Medical (Halle, Germany). Low molecular weight (LMW) chitosan, with a molecular weight range between 50 and 190 kDa, chitosan of 5 kDa molecular weight and GRS Taq DNA polymerase were purchased from Sigma Aldrich Chemicals (St. Louis, MO, USA). Chitosan of 20 kDa molecular weight was purchased from Gletham Life Sciences (Corsham, UK). Sodium tripolyphosphate (TPP) was obtained from Across Organics (Geel, Belgium). TripleXtractor used in RNA extraction was obtained from GRISP (Porto, Portugal). DMEM-F12 was purchased from GIBCO (Waltham, MA, USA). MEM-α was purchased from HyClone classical media (Boston, MA, USA). Sodium bicarbonate was obtained from MP Biomedicals (Santa Ana, CA, USA). Agarose and GreenSafe was obtained from NZYtech (Lisbon, Portugal).

#### 2.2.1. pDNA Amplification and Purification

The mutated E7 gene was cloned into pMC.CMV-MCS-EF1-GFP-SV40polyA parental minicircle cloning vector, next to CMV7 promoter, as described previously [13]. Thus, the plasmid DNA containing the mutated E7 gene (pDNA-E7mut) was amplified in ZYCY10P3S2T *Escherichia coli* host strain. Briefly, the host strain was inoculated in LB-agar petri dishes with kanamycin (50 mg/mL) and it was left overnight at 37 °C. After this incubation phase in solid medium, some colonies were transferred to a 250 mL Erlenmeyer containing 62.5 mL of pre-fermentation liquid medium, Terrific Broth (20 g/L of tryptone; 24 g/L of yeast extract; 4 mL/L of glycerol; 0.017 M KH_2_PO_4_, and 0.072 M K_2_HPO_4_, pH 7.0). The Erlenmeyer was placed in the orbital shaker at 42 °C and 250 rpm until the optic density (OD_600nm_) at 600 nm reaches 2.6. At this moment, a certain volume of pre-fermentation medium was transferred to a 1 L Erlenmeyer containing 250 mL of Terrific Broth medium to start the fermentation with an OD_600nm_ of 0.2. The Erlenmeyer was placed in the orbital shaker under previous conditions and left overnight (approximately for 12 h) until the bacterial growth reached the late log phase (OD_600nm_ ~ 9). Concluding this process, the bacterial growth was suspended by collecting the cells through centrifugation at 4500 g for 10 min at 4 °C, and the resultant pellets were stored at −2 °C. Afterward, the pDNA-E7mut was purified by NZYMaxiprep kit (Nzytech, Lisbon, Portugal), according to the supplier’s protocol.

#### 2.2.2. Agarose Gel Electrophoresis

Along the work, all the agarose gel electrophoresis were prepared at 1% (*w*/*v*), by adding 1 g of agarose to 100 mL of 1× TAE buffer (40 mM Tris base, 20 nM acetic acid, 1 mM EDTA at pH 8.0) and stained with 1.2 μL of Green Safe from NZYTech (Lisbon, Portugal). Electrophoresis was carried-out for 40 min at 150 Volts and then the gels were imaged using ultraviolet (UV) light, by the Uvitec Fire-Reader system (UVITEC, UK).

#### 2.2.3. Preparation of Chitosan/TPP/pDNA Nanoparticles

A stock solution (1 mg/mL) of each Chitosan polymer was prepared by resuspending the respective chitosan powder in sodium acetate buffer (0.1 M, pH 4.6), and a stock solution (1 mg/mL) of TPP was resuspended in the same buffer. In this work, four chitosan polymers were used with different molecular weights: namely the chitosan of HMW (200–500 kDa), LMW (50–190 kDa), 20 and 5 kDa. These solutions were stored at room temperature until use. The formulation of chitosan nanoparticles was performed by exploring the negatively charged TPP and the positively charged chitosan. To determine the minimum and maximum level of each input, some preliminary studies were performed, exploring different techniques of nanoparticle formulation (complex coacervation and ionotropic gelation) and varying the chitosan and TPP concentrations between 0.1 and 1 mg/mL. Based on these studies, the levels (−1; 0; +1) of the input chitosan concentration were defined to 0.2, 0.5 and 0.8 mg/mL, while 0.25, 0.5 and 0.75 mg/mL were identified for the TPP concentration. The pDNA-E7mut was added to the TPP solutions to attain a final concentration of 20 µg/mL. Nanoparticles were formulated by adding, drop by drop, 100 μL of the TPP/pDNA solution to 400 μL of chitosan solution under vortexing for 1 min, with a volume ratio of chitosan and TPP + pDNA solutions of 4:1. Chitosan/TPP/pDNA-E7mut nanoparticles were left for 30 min at room temperature for stabilization, being subsequently centrifuged at 10,000× *g* for 10 min at 4 °C.

#### 2.2.4. Design of Experiments (DoE)

DoE was used to optimize the formulation of chitosan/TPP/pDNA nanoparticles of each chitosan polymer, minimizing the nanoparticles size, while maximizing the zeta potential and maintaining a suitable PDI (outputs). Concerning that, a CCF was applied. Two factors were considered as inputs, namely the chitosan and TPP concentration. These inputs were studied at three levels (−1; 0; +1) and the range was defined according to preliminary studies. Statistical analysis was performed, using Design-Expert version 11. The generalized second-order polynomial model equation used in the response surface analysis is presented below (Equation (1)):(1)$$Y=\beta_{0}+\beta_{1}X_{1}+\beta_{2}X_{2}+\beta_{11}\;X_{1}^{2}+\beta_{22}\;X_{2}^{2}+\beta_{12}\;X_{1}X_{2}$$

After the analysis of the model suitability applied in DoEs, the optimal point to formulate nanoparticles with each chitosan polymer was determined and validated. The following studies were performed using input conditions determined in the optimal points.

#### 2.2.5. Characterization of Chitosan/TPP/pDNA Nanoparticles

Nanoparticles were characterized to obtain the respective DoE outputs (size, potential zeta and PDI) and to determine their performance as DNA delivery systems. The average size, PDI and zeta potential measurements were performed by Dynamic Light Scattering (DLS) at 25 °C using a Zetasizer Nano ZS equipment (Malvern Instruments, UK) and the Malvern Zetasizer software v6.34. Briefly, the average size and PDI were evaluated by mixing 500 µL of nanoparticles solution and 500 µL of ultrapure water and placed in a disposable cell. To evaluate the zeta potential, nanoparticles were centrifuged at 10,000× *g* for 10 min at 4 °C and resuspended in MES buffer (0.01 M, pH 6,2). Each parameter was measured three times from three independent samples (*n* = 3), except the results of runs proposed by the DoE that are resultant from three measurements of each sample (*n* = 1). The DoE proposes to perform three runs (central points) under the same conditions (*n* = 3) to assess the model reproducibility.

Scanning electron microscopy (SEM) was used to evaluate the chitosan nanoparticles morphology obtained with optimal point conditions established by DoE to each polymer. Nanoparticles were centrifuged at 10,000× *g* for 10 min at 4 °C, then resuspend in a PBS solution for cleaning impurities. Once again, nanoparticles were centrifuged under previous conditions and resuspended in 40 µL of a solution containing tungsten. This solution was then placed in a round-shaped coverslip and dried overnight at room temperature. The dry samples were sputter-coated with gold using an EMitech K550 (London, UK) sputter coater. A Hitachi S-2700 (Tokyo, Japan) scanning electron microscope, with an accelerating voltage of 20 kV at various magnifications, was used.

#### 2.2.6. Fourier Transform Infrared Spectroscopy

Fourier transform infrared spectroscopy (FTIR) was applied to evaluate the chemical structure of chitosan nanoparticles. The spectra were acquired using a Nicolet iS10 FTIR spectrophotometer (Thermo Scientific, Waltham, MA, USA) with an average of 120 scans, a spectral width ranging from 4000 to 800 cm^−1^ and a spectral resolution of 32 cm^−1^. The spectra of isolated chitosan, pDNA and TPP samples were acquired for comparative analysis.

#### 2.2.7. Stability Assays

Two different stability assays were performed with the formulated nanoparticles. The first assay aimed to assess the nanoparticle’s colloidal stability over time, storing them in the formulation buffer and at different temperatures [32,33,34]. For that, nanoparticles were stored in sodium acetate buffer, at room temperature or 4 °C. Nanoparticle characteristics, such as size, PDI and zeta potential, were evaluated throughout the time at 1 h, 1 day, 1 week and 1 month. The statistical analysis was performed with two-way ANOVA followed by Tukey test (GraphPad Prism 9 software, San Diego, CA, USA). The second assay aimed to evaluate the nanoparticle’s stability upon their incubation in the cell culture medium, with and without FBS supplementation, to mimic the in vitro cell transfection conditions and extracellular environment of human body, to ensure the success of cellular transfection of these chitosan nanoparticles [13,35]. Thus, nanoparticles were centrifuged and resuspended in 50 μL of DMEM-F12 or MEM alpha medium supplemented with 10% FBS or without this supplementation. Nanoparticles were further incubated at 37 °C for 6 h and the pDNA release and degradation were monitored by 1% agarose gel electrophoresis. Decomplexation/degradation studies were performed to ensure that pDNA remained intact inside the delivery systems. The nanoparticles were incubated at 60 °C in the presence of heparin for 1 h [36].

#### 2.2.8. Cell Culture and In Vitro Transfection Studies

The transfection studies were performed using human fibroblast cells (ATCC^®^ PCS-201-012™) and murine immature dendritic JAWSII cells (ATCC^®^ CRL-11904™). The human fibroblast cells were grown in DMEM/Ham’s F-12 Nutrient Mixture (DMEM-F12) and JAWSII cells were grown in MEMα supplemented with GM-CSF (5 ng/mL). Both media were supplemented with 10% (*v*/*v*) fetal bovine serum (FBS) and 1% (*v*/*v*) of a mixture of penicillin (100 μg/mL) and streptomycin (100 μg/mL). Cells were incubated at 37 °C in a humidified atmosphere containing 5% CO_2_. Cells were seeded in a complete medium and 24 h before transfections, this medium was replaced by medium without supplements to promote transfection. Cells were transfected when 50–60% confluence was achieved, by adding to each well different nanoparticles resuspended in the medium without supplements. After 6 h of transfection, the medium was changed to fresh complete medium and cells were allowed to grow for different times, depending on the intended experiment.

#### 2.2.9. Cytotoxicity Assays

To evaluate the nanoparticle’s biocompatibility, a resazurin assay was used. This assay was applied in two different cell lines, hFibro and JAWSII cells. Cells were seeded in 96-well plates with a density of 1 × 10^4^ cells/well. Transfections with different chitosan systems and naked pDNA-E7mut were performed for 48 and 72 h. After, the culture medium was discarded and 100 µL of fresh complete medium and 20 µL of resazurin 0.1% (*w*/*w*) were added to each well. Cells were incubated in the dark over 4 h at 37 °C in a humidified atmosphere containing 5% CO_2_. Finally, the resorufin fluorescence was measured in a spectrofluorometer (SpectraMAX^®^ GeminiTM EM, Molecular Devices, San Jose, CA, USA) by defining an excitation wavelength of 544 nm and an emission wavelength of 590 nm. Non-transfected cells were used as negative control and cells treated with 70% ethanol were used as the positive control. The statistical analysis was performed with two-way ANOVA followed by Tukey test (GraphPad Prism 9 software, San Diego, CA, USA).

## 3. Results and Discussion

### 3.1. DoE Inputs

Chitosan has been a widely used polymer in the formation of gene delivery systems. Several methods have been used to prepare chitosan delivery systems, and most of them consist of the interaction of positively charged chitosan and a negatively charged linker, such as DNA or TPP [15,37,38,39,40,41,42,43]. In the beginning of this work, two methods were used for the preparation of the delivery systems: complex coacervation and ionotropic gelation. One of the most important factors for the formation of these complexes is the chitosan/TPP volume ratio. This factor affects the complex properties, like size, PDI, zeta potential and morphology. Considering previous studies, the chitosan/TPP volume ratio that resulted in complexes with good physicochemical properties was 4:1 [22,37,38]. Despite the chitosan/TPP volume ratio and concentration of respective solutions, other factors can affect the characteristics of these delivery systems, such as the molecular weight of chitosan polymers. Considering these key factors and aiming to an effective and efficient production of delivery systems, the influence of molecular weight was also evaluated in this work, optimizing by DoE the respective nanoformulations with the chitosan of HMW (200–500 kDa), LMW (50–190 kDa), 20 and 5 kDa.

To define the formulation technique and the input levels of the chitosan and TPP concentration, some preliminary studies were performed exploring the range of 0.1–1 mg/mL. The delivery systems prepared by the complex coacervation technique showed aggregation, while the ionotropic gelation technique led to nanodelivery systems with acceptable characteristics. These chitosan nanoparticles presented an average size below 500 nm, a portion of these carriers had positive charges and no aggregation was obtained for most of the concentrations tested. Given this behavior, ionotropic gelation was the selected formulation technique to proceed the work and input levels were defined to 0.2–0.8 mg/mL for chitosan polymers concentration and to 0.25–0.75 mg/mL for TPP concentrations. The chitosan/TPP volume ratio and pDNA concentration were kept constant along the study, being 4:1 and 20 μg/mL, respectively.

### 3.2. Model Application and Analysis

After defining the DoE inputs, a three-level CCF design was applied to achieve the optimal nanoparticle formulation. This particular design is adequate for the present work because it does not consider points outside of the ranges established for the inputs [31]. Considering the results attained in preliminary studies, unsatisfactory results were expected using inputs outside the ranges, such as particle aggregation. In addition, the CCF model also allows the reduction of the number of experiments required to identify the optimal nanoparticles formulation. In Table 1, all runs proposed by the DoE tool are presented to the four chitosan polymers, showing the concentrations used on the nanoparticles formulation and the respective outputs. The model reproducibility was evaluated considering three central points (*n* = 3) to be performed under the same conditions, marked in grey in Table 1. The zeta potential, size and PDI outputs were determined by DLS. To avoid results inconsistence, all data shown in Table 1 represents the mean of three measurements for each output (run). 

In general, the data demonstrate that all experiments allowed to formulate nanoparticles with some variations of physicochemical properties. The size and zeta potential of chitosan nanoparticles are mainly dependent on the molecular weight of the used polymer and in less extension, the concentration of the polymer and TPP crosslinker considered. In this way, the size of chitosan nanoparticles decreased when the molecular weight of chitosan polymers is lower, whereby the HMW chitosan led to the highest particle size of 249 nm and the 5 kDa chitosan allowed the production of particles with the lowest size of 74.82 nm. According to some studies, this behavior in the chitosan nanoparticles size can be explained by the increase of the viscosity and the decrease of the solubility when a HMW polymer is used [44,45]. For instance, when the molecular weight increases, the size and zeta potential of delivery systems also increases. Some works describe that HMW chitosan polymers can be more stable than LMW chitosan polymers, while LMW chitosan can release the DNA at higher extent than the HMW chitosan, which can be useful for gene expression if nanosystems are inside the cells [15,44]. The chitosan concentration differently affects the nanoparticles size in the four used polymers (Table 1). The HMW and LMW chitosan polymers formulate small nanoparticles when the chitosan concentration is low. The 5 and 20 kDa chitosan polymers led to small nanoparticle sizes for intermediate and high concentrations, respectively. These results suggest that chitosan polymer concentration may depend on the molecular weight of the used polymer, since polymers with lower MW require higher concentrations to get the same cross-linking density with the TPP/pDNA than chitosan polymers with higher MW [23]. Looking at Table 1, the influence of TPP concentration on nanoparticles size is also variable in function of different chitosan polymers. For the HMW chitosan, the nanoparticles size is smaller when the TPP concentration is low. The LMW and 20 kDa chitosan nanoparticles presented lower mean diameters when higher TPP concentrations were used in the formulations. As already observed before, the nanoparticles size is dependent on a strong relation between the molecular size and concentration of chitosan and the contribution offered by the TPP concentration to the pDNA solution. The lower the molecular weight of the polymer used, the greater amount of this polymer and TPP will be needed to establish crosslinking with all chitosan chains present in solution [22].

On the other hand, the zeta potential output increased with the increase of the molecular weight and the concentration of the chitosan polymer, since the presence of positive charges also increases [42]. With the TPP concentration increase, the zeta potential of nanoparticles decreased in all polymers. This behavior is expected because TPP increases the negative charge of solution. In general, the PDI output was below 0.4, which indicates that the formulations present some homogeneity [46]. Its variation is inconstant for the different polymers, although it is possible to verify a downward trend when the concentrations of TPP increase and the chitosan concentration remains constant. This slight decrease of the nanoparticles’ PDI can be justified by the ionotropic gelation technique used, which consists of adding the TPP/pDNA solution drop by drop in the chitosan solution, under constant agitation, resulting in nanoparticles of comparable size [22]. It is noteworthy that the HMW chitosan provided nanoparticles with the highest zeta potential (+32.0 mV) and the 5 kDa chitosan enabled nanoparticles with the lowest size (74.82 nm), both with suitable PDIs.

Table 2 presents multiple regression equations and the surface response model chosen for each output of each chitosan polymer. These equations are provided by the Design Expert analysis and indicate the level of the outputs as a function of different inputs, where the signal behind each factor represents a positive or a negative effect in the outputs [31]. Factor A represents the chitosan concentration and factor B represents the TPP concentration. Concerning the nanoparticle size output, Table 2 shows that the increment of the chitosan concentration input had a positive effect in the LMW and 5 kDa chitosan polymers. On the other hand, the increment of the TPP concentration negatively influenced the size of nanoparticles prepared using almost all polymers. On the one hand, this behavior of each isolated output shows that the increase of chitosan concentration implies more positive chain polymers to easily condense the same amount of negatively charged molecules (pDNA and TPP) [23], while on the other hand, the increase of TPP concentration weakens the condensation power of the same amount of chitosan polymer. In addition, for the PDI and zeta potential outputs, it is evident that the chitosan concentration input induced a positive effect in almost all chitosan polymers and the TPP concentration input had a negative effect. This negative impact of the TPP can be explained by the increase of negative charges of this cross-linker in the formulation process. In addition, the increment of the TPP concentration should be controlled because it can be related to aggregation phenomena [38,40]. As evidenced in Table 2, when the surface response model chosen is quadratic, the respective output regression equation presents more information related to the input interactions (AB) and the input effect on itself (A^2^ or B^2^), while the linear model only presents information of each isolated input.

The data represented in Table 3 and Table 4 correspond to the statistical coefficients and the analysis of the variance (ANOVA) of four DoE, respectively, and allow to assess the goodness of fit. Based on the parameters of Table 3 and Table 4, it is possible to evaluate the significance and adequacy of the used models, and to understand if these experiments are valid and fit the data through the statistical models generated. In order to evaluate if the model has high significance and the fitness of the output statistical model to the data, the coefficient of determination (R^2^) has to be close to 1 [47]. In general, the R^2^ values obtained in all outputs of studied chitosan polymers were around 0.9, suggesting that all models fit the data. However, the values of R^2^ for PDI output of the 20 and 5 kDa chitosan polymers were 0.62 and 0.82, respectively, which indicates the polynomial model is not a good predictor of this response. This result can be related with the absence of a tendency of this response observed in Table 1 (as it was already previously described), but the obtained results (PDI below 0.4) are suitable since we intend to get homogeneous samples [46]. Adjusted R^2^ values indicate if the theoretical values adjust to the experimental data and should not be lower than 0.2 from its R^2^ to indicate that the sample size is adequate for the model [28]. Results from Table 3 show that all values of adjusted R^2^ follow the same tendency of R^2^ and never lower more than 0.09 in comparison to its R^2^. The predicted R^2^ gives information regarding the predictive power of these models. As the results of predicted R^2^ presented in Table 3 are all positive, it confirms the model’s capacity in predicting new data. Lastly, adequate precision values are presented in Table 3, measuring the signal due to the noise ratio. These values must be greater than 4 to consider that the model provides an adequate signal and can be used to navigate the design space [26]. All adequate precision results were higher than 7.6, confirming an adequate signal-to-noise ratio. The study of all these coefficients shows that the quadratic and linear models chosen in each output (Table 2) were adequate for the statistical analysis of these results.

ANOVA analysis was performed to further prove the validity of each DoE. The results shown in Table 4 represent the model significance for each output and the correspondent lack of fit. To have a good valid model, a significant value for the model (*p*-value < 0.05) and a non-significant value for the lack of fit (*p*-value > 0.05) is necessary [48]. When the model respects these parameters, it indicates that the model data are significant and fit. According to the *p*-values presented in Table 4, all the model values are significant and do not present a significant lack of fit. With this data analysis, it can be confirmed that the statistical model is good and valid for all outputs. In addition, Table 4 also shows that inputs have a significant influence on all outputs, except for the chitosan concentration in the nanoparticles prepared using HMW and 5 kDa polymers and in the PDI of the 20 kDa polymer.

After the statistical analysis of the coefficients, the analysis of the variance and validating the statistical models of the DoE, the Design-Expert software predicted the optimal points to the four chitosan polymers. Table 5 represents the concentration of each factor A and B (chitosan concentration and TPP concentration) that is required to obtain the optimal formulation of nanoparticles with each polymer, as well as predicted outputs and a range of values for the validation of each output (95% of confidence interval). The concentration of each factor to perform the optimal formulation was predicted, aiming at chitosan nanoparticles with the smallest size, highest zeta potential and a PDI below 0.4. These predicted values are based on inputs and experiments previously performed, and as expected, each predicted optimal point has different input conditions since results of Table 1 are strongly dependent on the applied polymer. Thus, using the chitosan concentration of 0.51 mg/mL and TPP concentration of 0.25 mg/mL with the HMW chitosan, the outputs predict nanoparticles with size of 125.28 nm, zeta potential of +30.1 mV and PDI of 0.228; using the chitosan concentration of 0.2 mg/mL and TPP concentration of 0.41 mg/mL with the LMW chitosan, the outputs predict nanoparticles with size of 94.25 nm, zeta potential of +16.1 mV and PDI of 0.256; using the chitosan concentration of 0.8 mg/mL and TPP concentration of 0.34 mg/mL with the 20 kDa chitosan, the outputs predict nanoparticles with size of 104.31 nm, zeta potential of +21.9 mV and PDI of 0.299 and using the chitosan concentration of 0.56 mg/mL and TPP concentration of 0.41 mg/mL with the 5 kDa chitosan, the outputs predict nanoparticles with size of 81.27 nm, zeta potential of +20.4 mV and PDI of 0.257. These optimal formulations were performed, and the mean of each obtained output results from three independent experiments, being the data presented in Table 5. All polymers achieved a result within the confidence interval provided by the Design-Expert software, concluding that all the outputs are valid.

Comparing the results of the four optimal points, the HMW chitosan resulted in nanoparticles with the highest zeta potential (+26.8 mV) but also the largest size (139.20 nm), as expected. However, the 20 and 5 kDa chitosan polymers gave rise to nanoparticles of reasonable zeta potential and the size was smaller than the HMW chitosan. The 5 kDa chitosan polymer led to nanoparticles with the smaller size (81.66 nm) of four polymers. The lowest zeta potential corresponded to the LMW chitosan (15.9 mV), although the size was equivalent to the nanoparticles obtained using the 20 kDa chitosan. The PDI achieved for all polymers was about the same (0.22–0.26), confirming that all formulations were homogeneous. These results are consistent with previous works. Huang and co-workers achieved LMW chitosan nanoparticles with size of 94 nm, zeta potential of +5.71 mV and PDI of 0.278 [33]. Although the size of LMW chitosan nanoparticles in this experiment increased by 4 nm, the PDI decreased and the zeta potential is much more positive (+15.9 mV) than the nanoparticles presented by the previous work. In another work using LMW chitosan, nanoparticles with average size of 108 nm and a zeta potential of +6 mV were formulated [49]. The nanoparticles obtained in the present work have a smaller size (97.82 nm) and a bigger zeta potential (+15.9 mV), which can be related to the different used formulation method or the application of the DoE tool. While, nanoparticles prepared with HMW chitosan presented an average size of 181 nm and zeta potential of 22.2 mV [50]. Özbaş-Turan and co-workers also tested the formulation of nanoparticles with HMW chitosan, but without the DoE and the resultant size remained above 190 nm [51]. Supported by DoE tool, nanoparticles obtained in the present work with the HMW chitosan are smaller (139.2 nm) and also more positively charged (+26.8 mV) than the presented formerly. Thus, the present work shows that DoE can be universally applied to optimize the formulation of nanoparticles to deliver DNA by using whatever material compared to the common random experiment approach [25]. Deng and co-workers produced nanoparticles based on 5 kDa chitosan with size of 144.8 nm and a zeta potential of +28 mV [38]. Their studies explored chitosan/hyaluronic acid to form nanoparticles, while in our study, chitosan/TPP was used, which can explain the difference in size, given the TPP is a stronger cross-linker, creating smaller nanoparticles (81.66 nm). However, the use of TPP can be responsible for the decrease of zeta potential (19.8 mV) due to the increment of the global negative charge of pDNA solution.

This study of experimental design for the optimization of the formulation of different chitosan polymers was successfully achieved. It is possible to notice differences in the outputs, between chitosan polymers with different molecular weights, mainly in terms of nanoparticles’ sizes and charges, both parameters being higher for longer polymer chains. Additionally, the concentration of chitosan and TPP have a strong influence on these outputs. In addition, the nanoformulations obtained with the four polymers analyzed by SEM showed spherical or oval morphology (data not shown). This study demonstrates that it is possible to form nanoparticles with chitosan polymers of different molecular weight and improve their performance by choosing the right chitosan and TPP concentrations. This optimization process becomes extremely relevant when developing a delivery system for cellular transfection, aiming for therapeutic gene expression. The cellular uptake/internalization is favored by carriers exhibiting a spherical morphology, small size (˂200 nm), monodispersity and high positive surface charges [52,53,54].

Therefore, this kind of DoE study enabled us to have a deep control over the characteristics of the formed delivery systems, thus, potentiating the desired therapeutic response.

### 3.3. Fourier Transform Infrared Spectroscopy

The FTIR technique was performed to evaluate the chemical structure of chitosan, as well as the possible interactions between pDNA and TPP. Therefore, the FTIR spectra of pDNA, TPP, HMW, LMW, 20 and 5 kDa chitosan and chitosan nanoparticles are shown in Figure 1, which reveals the fundamental absorption bands. Overall, the spectra of chitosan with different molecular weights exhibited characteristic peaks with strong similarity. The strong bands in the region 3396–3144 cm^−1^ can be attributed to N-H and O-H stretching vibrations, as well as the intramolecular hydrogen bonds. The absorptions bands at 2894–2880 cm^−1^ correspond to C-H symmetric and asymmetric stretching vibrations. Bands in the region 1608–1547 cm^−1^ are related to residual N-acetyl groups in chitosan and bands between 1082 and 1074 cm^−1^ are attributed to the asymmetric stretching of the C–O–C bridge [55]. Similar behavior has already been reported by previous studies, in which functional groups present in chitosan with different molecular weights were assessed [56,57,58]. Concerning the pDNA spectrum, the peaks in the region of 1700–1500 cm^−1^ correspond to the nitrogen bases. Furthermore, the peak at 1116 cm^−1^ can be assigned to the vibration of ribose (C-C sugar) and the peak at 944 cm^−1^ is indicative of the presence of pDNA [35]. TPP spectrum demonstrated characteristic bands at 1214 cm^−1^ (P-O stretching vibration), 1084 cm^−1^ (symmetric and asymmetric stretching vibrations in PO_2_ group), 973 cm^−1^ (symmetric and asymmetric stretching vibrations in PO_3_ group) and 908 cm^−1^ (asymmetric stretching of the P-O-P bridge) [59,60]. Regarding the spectra of chitosan/TPP/pDNA nanoparticles, a broad band in the region 3400–3200 cm^−1^ reveals enhanced intermolecular hydrogen bonding due to strong interaction bonding with TPP and pDNA. Furthermore, a new absorption peak appeared at 1414 cm^−1^, which is assigned to N-O-P stretching vibrations. This evidence indicates that TPP anions were crosslinked with the amine groups of chitosan to form complexes, and similar results were obtained in other works [59,61].

### 3.4. Stability Assays

After establishing the optimal points for the nanoparticle’s formulation with chitosan polymers with different molecular weights, it was important to evaluate the nanoparticles colloidal stability, when they are stored in the formulation buffer over time and at different temperatures [32,33,34]. Thus, the size, potential zeta and PDI of stored nanoparticles were measured after one hour, one day, one week and one month of storage in the sodium acetate buffer (0.1 M, pH 4.6), at 4 °C and room temperature. Analyzing the data in Figure 2, the size of nanoparticles presents absence or small changes over time (no more than 20 nm after one month) less evident at 4 °C for LMW, 20 and 5 kDa chitosans, always remaining near to optimal point results of the delivery systems obtained from the DoE. According to previous studies the chitosan/TPP nanoparticles only show small or moderate size changes after a month of storage in both temperatures [32,33,34]. These results suggest the storage conditions are suitable to avoid spontaneous aggregation, erosion of particles spherical shape or swelling the chitosan nanoparticles (mechanisms responsible for a strong increase in chitosan-TPP particle size from 400 to 900 nm after 7 days for instance), most probably by reducing the frequency of particle collisions at low temperatures [32]. The PDI of these systems exhibit some variability over time for the four polymers, being less variable with HMW and 5 kDa chitosans. However, the results indicate some homogeneity of nanoparticle size in each polymer over time since all PDI remained below 0.4. The zeta potential of nanoparticles at room temperature was significantly reduced in all nanosystems, and this decrease ranged between 20 and 35%, depending of the polymer. Moderate changes in the zeta potential were observed when particles prepared with HMW and LMW polymers were stored at 4 °C, while particles obtained with 20 and 5 kDa chitosan polymers did not show any significant changes over one month. This behavior of HMW (200–500 kDa) and LMW (50–190 kDa) can be related to the chemical breakdown in the polymeric network of the nanoparticles obtained with polymers which present chains with variable sizes into a specific range [46]. Previous studies demonstrate that chitosan/TPP nanoparticles, after one month in deionized water, changed drastically their characteristics compared to when they were prepared [62]. This set of results suggests that the nanosystems obtained with the 20 and 5 kDa chitosan polymers can be more stable than the ones formed with HMW and LMW and should be stored in the formulation buffer (acetate buffer (0.1 M, pH 4.6)) at 4 °C over time.

Furthermore, the nanoparticles stability was also evaluated following their incubation in the cell culture medium, with or without FBS supplementation, to simulate the in vitro cell transfection and in vivo extracellular conditions, to ensure the success of cellular transfection of these chitosan nanoparticles [13,35]. Thus, all chitosan nanoparticles were resuspended and incubated for 6 h at 37 °C in two different mediums, MEM-α (normally used for dendritic cell culture) and DMEM-F12 (normally used for human fibroblast culture) with and without supplements. The supplements 10% FBS and 1% penicillin-streptomycin were subsequently added. The stability test performed in medium with supplementation aimed to mimic the extracellular environment of the human body. After 6 h, an 1% agarose gel electrophoresis was performed to monitor if the pDNA was released and degraded during the incubation process. As demonstrated in Figure 3, for both media, the different chitosan nanoparticles remained stable since it was not possible to find pDNA at the first moment of resuspension (0 h) or after 6 h of the incubation process, due to the absence of nucleic acid bands in the electrophoretic mobility profile. It is noteworthy that the presence of FBS in the medium does not have any destabilizing effect on the nanoparticles, being a good indicator of the resilience of the nanoparticles in the extracellular environment [13,35]. The effect observed in the agarose gels of Figure 3C,D is due to the culture medium, since lane 2 corresponds only the sample medium and presents the same aspect. In addition, chitosan nanoparticles were also decomplexated/degraded with heparin and the electrophoresis confirmed the release of encapsulated pDNA and the absence of degradation (data not shown). These results suggest that chitosan nanoparticles can efficiently protect the pDNA vector and are suitable delivery systems for DNA vaccines [36].

Thus, considering the stability of chitosan nanoparticles, incubated in cell culture medium and in the formulation buffer over time, in vitro transfection studies are essential to evaluate the biocompatibility of chitosan nanoparticles.

### 3.5. Cytotoxicity Assays

A cytotoxicity assay was performed to evaluate the biocompatibility of formulated chitosan nanoparticles. The addition of resazurin to the transfected cells determines if the chitosan systems had any toxic effects on them, through the analysis of resorufin fluorescence produced [63]. As DNA vaccines require the expression of pDNA encoding antigens by antigen presenting cells (APCs) to induce immune responses, JAWSII cells were included in this study, which are known as difficult to transfect [64,65]. To guarantee the safety of this approach towards other cell lines, human fibroblast cells were selected since they are widely used in cell culture. Thus, the results on JAWSII and human fibroblast cells at 48 and 72 h after transfection with the different chitosan/TPP/pDNA systems and naked DNA are presented in Figure 4. For both cell lines, it was observed that the chitosan nanoparticles did not show toxic effects since the cell viability was always superior to 90%. Similar findings were described by Hallaj-Nezhadi and co-workers, where chitosan-DNA nanoparticles showed no cell toxicity in murine CT-26 colon carcinoma cells, exhibiting average cell viabilities over 90% [66]. Therefore, the formulated delivery systems are not toxic to cells at 48 and 72 h after transfection. Furthermore, no significant differences were observed in the cell viability profile between the different chitosan systems. In contrast, cells transfected with naked DNA (nkDNA) had a significant influence on cell viability when compared to non-transfected cells. This data demonstrates that the chitosan nanoparticles help in the delivery of the pDNA to the cells, avoiding the pDNA degradation and toxicity to the cells. These results suggest an improvement in biocompatibility when chitosan/TPP/pDNA systems are used compared to the delivery of the pDNA-E7mut vaccine alone.

## 4. Conclusions

Considering the health challenges of current times, DNA vaccines are a very promising technology for the years ahead. In the present work four chitosan polymers (HMW, LMW, 20 and 5 kDa) displaying different molecular weights were studied through a DoE tool. The properties (size, zeta potential, PDI, stability and cytotoxicity) of the chitosan/TPP/pDNA delivery systems were evaluated and the optimal formulation point was achieved successfully for each one of the four chitosan polymers. Despite the molecular weight differences, the optimal formulation point was achieved by the iteration between the concentrations of chitosan and TPP. However, each polymer has different characteristics, HMW chitosan produced the biggest delivery systems size (139.20 nm) and the highest zeta potential (+26.8 mV), 5 kDa chitosan created the smallest size delivery systems (81.66 nm) and a reasonable zeta potential (+19.8 mV), LMW and 20 kDa chitosans generated delivery systems with sizes of 97.82 and 97.67 nm and zeta potentials of +15.9 and 21.8 mV, respectively. Cytotoxicity assays reveal that none of the chitosans had a toxic effect on the human fibroblasts or JAWSII cells. Through stability studies it was possible to conclude that, at 4 °C, the delivery systems are more stable, after storage for a month in sodium acetate buffer. These results show that it is possible to use any of these polymers to create a good delivery system if the relation between the chitosan/TPP is optimized. DoE revealed to be a potent tool to unravel the most suitable carrier system. To evaluate the applicability of the formed chitosan-based systems in the DNA vaccines field, further studies are already being carried out to understand if the different polymers induce differences on the efficiency of cellular transfection with repercussions on gene expression. Results on this topic may highlight the different performance of the various chitosan systems, related with their properties, and hopefully, will be reported/published soon.

## Figures and Tables

**Figure 1 pharmaceutics-13-01369-f001:**
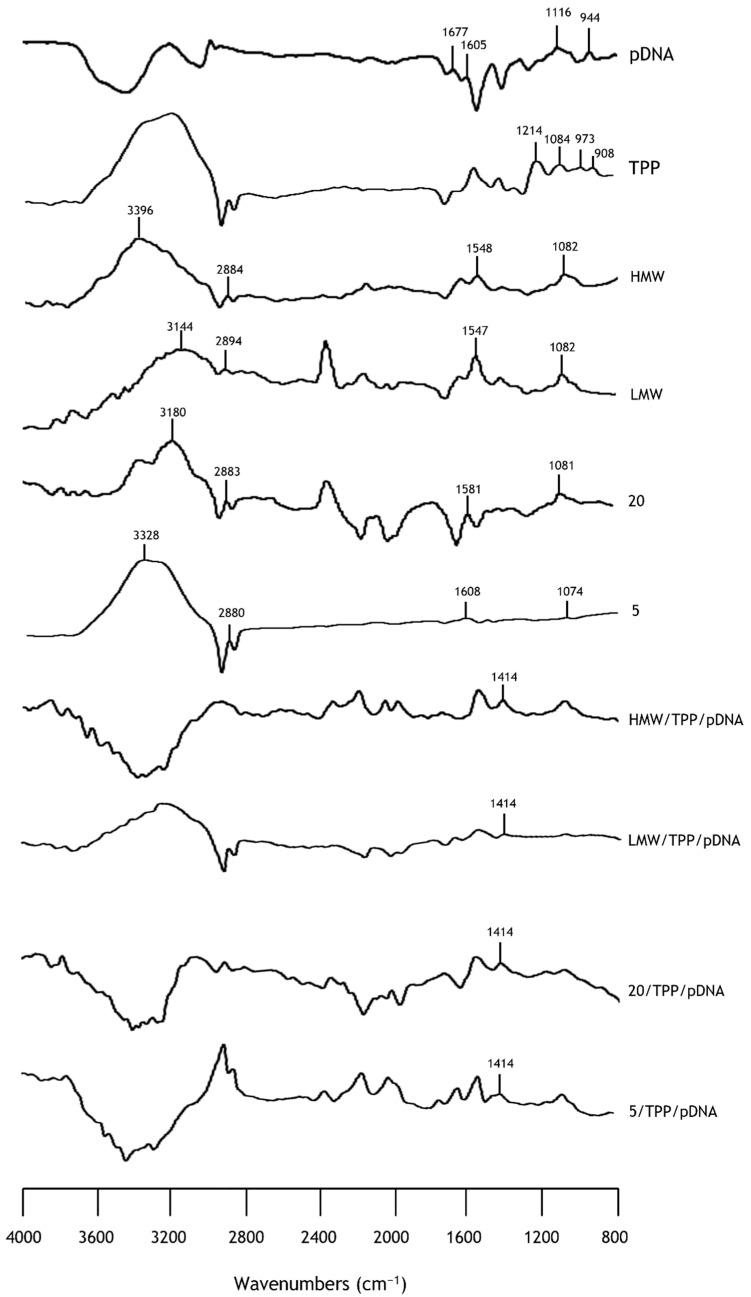
FTIR spectra (absorbance versus wavenumbers) of pDNA, TPP, chitosan with different molecular weights (HMW, LMW, 20 kDa, 5 kDa) and chitosan/TPP/pDNA nanoparticles.

**Figure 2 pharmaceutics-13-01369-f002:**
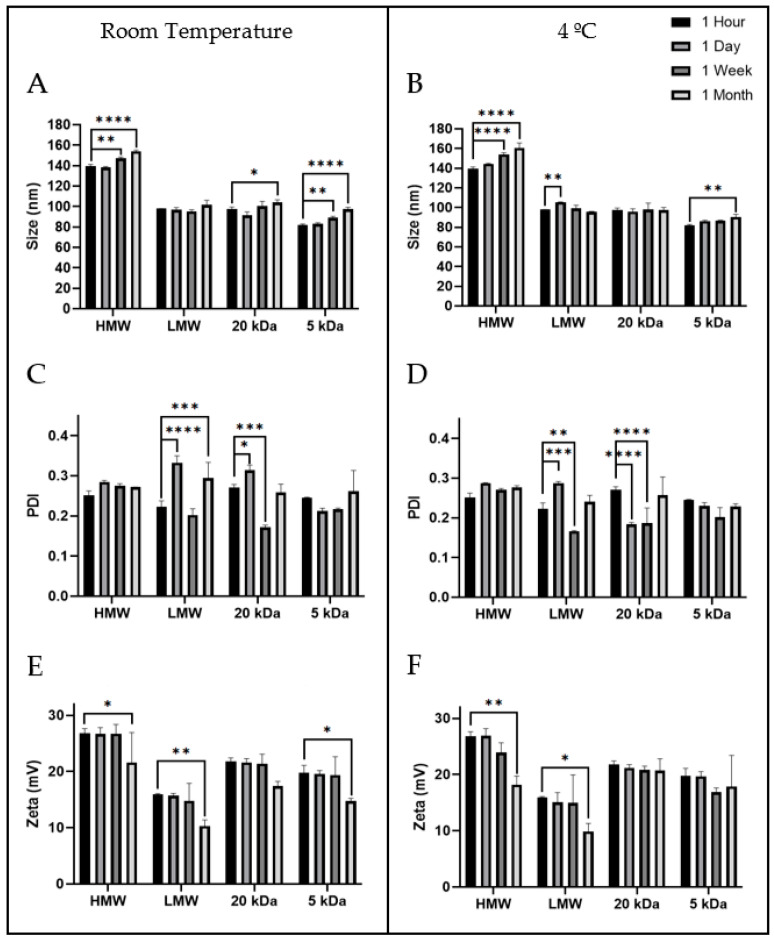
Stability assays of chitosan nanoparticles at 1 h, 1 day, 1 week and 1 month. (**A**) Chitosan nanoparticles size after storage at room temperature. (**B**) Chitosan nanoparticles size after storage at 4 °C. (**C**) Chitosan nanoparticles PDI after storage at room temperature. (**D**) Chitosan nanoparticles PDI after storage at 4 °C. (**E**) Chitosan nanoparticles zeta potential at room temperature. (**F**) Chitosan nanoparticles zeta potential at 4 °C. Data obtained from three independent samples (mean ± S.D., n = 3). The observed differences are statistically significant (* *p* < 0.05; ** *p* < 0.01; *** *p* < 0.001; **** *p* < 0.0001).

**Figure 3 pharmaceutics-13-01369-f003:**
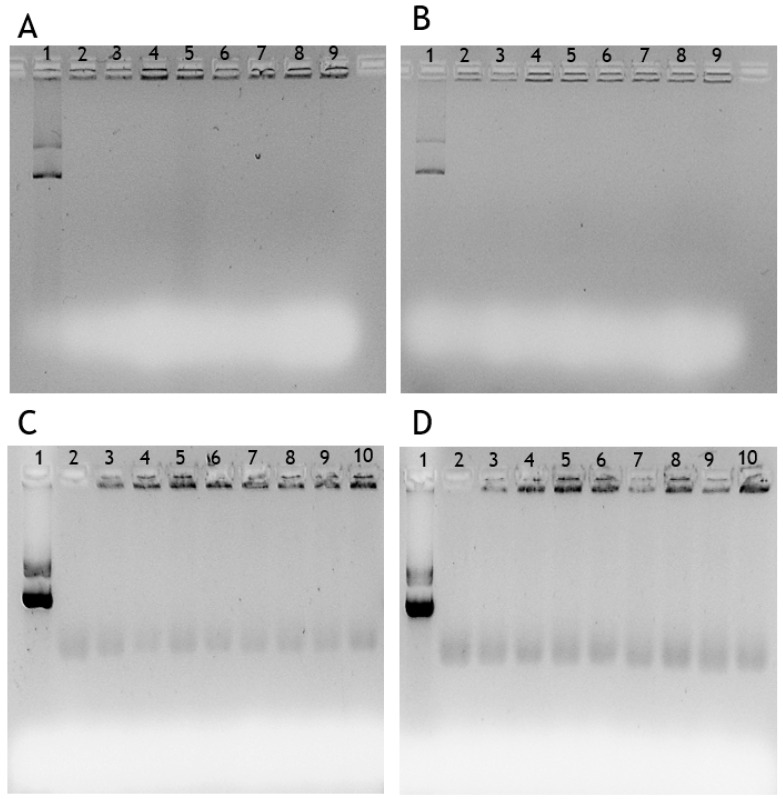
Analysis of nanoparticles’ stability in cellular incomplete medium by agarose gel electrophoresis. Evaluation of nanoparticles stability in MEM-α incomplete medium (**A**) and in DMEM-F12 incomplete medium (**B**) at 37 °C. Lane 1: pDNA E7 mut sample; lanes 2 and 3: nanoparticles obtained with HMW chitosan polymer incubated at 0 and 6 h, respectively; lanes 4 and 5: nanoparticles obtained with LMW chitosan polymer incubated at 0 and 6 h, respectively; lanes 6 and 7: nanoparticles obtained with 20 kDa chitosan polymer incubated at 0 and 6 h, respectively; lanes 8 and 9: nanoparticles obtained with 5 kDa chitosan polymer incubated at 0 and 6 h. Evaluation of nanoparticles stability in MEM-α complete medium (**C**) and in DMEM-F12 complete medium (**D**) at 37 °C. Lane 1: pDNA E7mut sample; lane 2: culture medium sample; lanes 3 and 4: nanoparticles obtained with HMW chitosan polymer incubated at 0 and 6 h, respectively; lanes 5 and 6: nanoparticles obtained with LMW chitosan polymer incubated at 0 and 6 h, respectively; lanes 7 and 8: nanoparticles obtained with 20 kDa chitosan polymer incubated at 0 and 6 h; lanes 9 and 10: nanoparticles obtained with 5 kDa chitosan polymer at 0 and 6 h.

**Figure 4 pharmaceutics-13-01369-f004:**
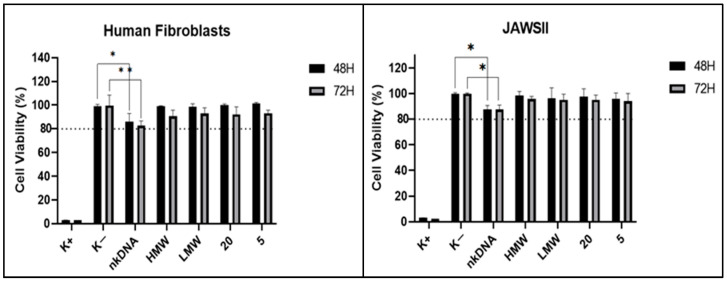
Cellular viability of JAWSII and human fibroblasts cells at 48 and 72 h after transfection with the different chitosan/TPP/pDNA systems and naked DNA. Cells treated with 70% ethanol were used as positive control (K+) and non-transfected cells were used as negative control (K−). The percent cell viability is expressed relative to the negative control. Data obtained from three independent measurements (mean ± S.D., n = 3). The difference between negative control and naked DNA is statistically significant (* *p* < 0.05; ** *p* < 0.01).

**Table 1 pharmaceutics-13-01369-t001:** Central composite design and response values for four chitosan polymers.

CHITOSAN POLYMERS	Chitosan Concentration (mg/mL)	Tpp Concentration (mg/mL)	Size (nm)	PDI	Zeta Potential (mV)
**HMW CHITOSAN**	0.20	0.25	115.0	0.280	20.57
	0.20	0.50	210	0.250	14.4
	0.20	0.75	249	0.220	9.82
	0.50	0.25	125.0	0.210	29.5
	0.50	0.50	156.0	0.220	25.6
	0.50	0.50	158.0	0.200	27.1
	0.50	0.50	167.0	0.210	25.6
	0.50	0.75	180.0	0.200	26.9
	0.80	0.25	133.0	0.330	30.3
	0.80	0.50	172.0	0.310	32.0
	0.80	0.75	216.0	0.270	28.0
**LMW CHITOSAN**	0.20	0.25	109.0	0.317	20.3
	0.20	0.50	88.05	0.237	14.4
	0.20	0.75	85.11	0.282	9.82
	0.50	0.25	106.6	0.384	20.9
	0.50	0.50	94.41	0.246	15.1
	0.50	0.50	98.00	0.265	15.9
	0.50	0.50	96.39	0.274	15.9
	0.50	0.75	96.81	0.228	13.6
	0.80	0.25	128.1	0.450	27.0
	0.80	0.50	114.1	0.289	24.0
	0.80	0.75	108.0	0.294	21.6
**20 KDA CHITOSAN**	0.20	0.25	141.5	0.324	17.5
	0.20	0.50	121.0	0.305	15.8
	0.20	0.75	116.0	0.290	10.1
	0.50	0.25	110.8	0.340	20.7
	0.50	0.50	106.1	0.336	17.1
	0.50	0.50	102.0	0.267	18.9
	0.50	0.50	102.6	0.359	18.1
	0.50	0.75	86.15	0.201	14.1
	0.80	0.25	107.0	0.331	21.9
	0.80	0.50	97.95	0.250	19.6
	0.80	0.75	92.54	0.177	17.0
**5 KDA CHITOSAN**	0.20	0.25	83.05	0.256	19.1
	0.20	0.50	84.41	0.209	17.5
	0.20	0.75	109.1	0.159	13.0
	0.50	0.25	91.28	0.262	21.8
	0.50	0.50	77.68	0.217	18.7
	0.50	0.50	77.91	0.235	17.5
	0.50	0.50	74.82	0.196	18.4
	0.50	0.75	80.11	0.186	15.2
	0.80	0.25	110.4	0.365	23.7
	0.80	0.50	90.83	0.240	23.2
	0.80	0.75	82.21	0.215	18.6

**Table 2 pharmaceutics-13-01369-t002:** Coded multiple regression equations for each output assessed in the chitosan nanosystems formulation.

Chitosan Polymers	Output	Multiple Regression Equations	Surface Response Model
**HMW** **CHITOSAN**	Size	+171.00 − 8.83 A + 45.33 B	Linear
	PDI	+0.21 + 0.027 A − 0.022 B + 0.070 A^2^ − 5.000 E − 003 B^2^	Quadratic
	Zeta potential	+26.76 + 7.59 A − 2.61 B + 2.11 AB − 4.55 A^2^ + 0.45 B^2^	Quadratic
**LMW** **CHITOSAN**	Size	+96.05 + 11.34 A − 8.96 B + 0.95 AB + 5.36 A^2^ + 5.99 B^2^	Quadratic
	PDI	+0.26 + 0.033 A − 0.058 B − 0.030 AB + 0.016 A^2^ + 0.059 B^2^	Quadratic
	Zeta potential	+15.87 + 4.68 A − 3.86 B + 1.27 AB + 2.97 A^2^ + 1.02 B^2^	Quadratic
**20 KDA** **CHITOSAN**	Size	+101.49 − 13.50 A − 10.77 B + 2.76 AB + 11.11 A^2^ + 0.11 B^2^	Quadratic
	PDI	+0.29 − 0.027 A − 0.054 B	Linear
	Zeta potential	+17.35 + 2.52 A − 3.15 B	Linear
**5 KDA** **CHITOSAN**	Size	+77.26 + 2.15 A − 3.34 B − 12.06 AB + 9.67 A^2^ + 8.10 B^2^	Quadratic
	PDI	+0.23 +0.033 A − 0.054 B	Linear
	Zeta potential	+18.79 + 2.65 A − 2.97 B	Linear

**Table 3 pharmaceutics-13-01369-t003:** Statistical coefficients obtained by the DoE for all chitosan polymers.

Chitosan Polymers	Output	R2	R2 Adjusted	R2 Predicted	Adequate Precision
**HMW** **CHITOSAN**	Size	0.7610	0.7013	0.4895	9.255
	PDI	0.9494	0.8988	0.5463	12.978
	Zeta potential	0.9637	0.9274	0.6932	14.865
**LMW** **CHITOSAN**	Size	0.9577	0.9154	0.5900	15.212
	PDI	0.9484	0.8969	0.5789	13.512
	Zeta potential	0.9945	0.9891	0.9686	43.376
**20 KDA** **CHITOSAN**	Size	0.9598	0.9196	0.7182	15.572
	PDI	0.6282	0.5352	0.3147	7.695
	Zeta potential	0.9229	0.9036	0.8315	21.500
**5 KDA** **CHITOSAN**	Size	0.9741	0.9482	0.7756	18.732
	PDI	0.8201	0.7752	0.5989	12.971
	Zeta potential	0.9249	0.9061	0.8681	21.915

**Table 4 pharmaceutics-13-01369-t004:** ANOVA *p*-values for central composite design for the formulation of chitosan nanosystems.

Chitosan Polymers	Source	Size	PDI	Zeta Potential
**HMW** **CHITOSAN**	Model	0.0033	0.0030	0.0013
	A	0.3626	0.0066	0.0002
	B	0.0011	0.0150	0.0184
	AB	-----	1.0000	0.0720
	A^2^	-----	0.0006	0.0114
	B^2^	-----	0.6093	0.7132
	Lack of fit	0.0504	0.2676	0.1276
**LMW** **CHITOSAN**	Model	0.0019	0.0031	<0.0001
	A	0.0006	0.0131	<0.0001
	B	0.0017	0.0012	<0.0001
	AB	0.6225	0.0365	0.0050
	A^2^	0.0649	0.2788	0.0003
	B^2^	0.0462	0.0069	0.0288
	Lack of fit	0.1448	0.2565	0.4143
**20 KDA** **CHITOSAN**	Model	0.0017	0.0191	<0.0001
	A	0.0006	0.1431	0.0003
	B	0.0016	0.0109	<0.0001
	AB	0.2509	-----	-----
	A^2^	0.0088	-----	-----
	B^2^	0.9697	-----	-----
	Lack of fit	0.1582	0.7251	0.4872
**5 KDA** **CHITOSAN**	Model	0.0006	0.0010	<0.0001
	A	0.0944	0.0140	0.0002
	B	0.0240	0.0009	<0.0001
	AB	0.0002	-----	-----
	A^2^	0.0018	-----	-----
	B^2^	0.0039	-----	-----
	Lack of fit	0.2601	0.3772	0.2739

**Table 5 pharmaceutics-13-01369-t005:** Optimal points for the four chitosan polymers.

Chitosan Polymers	Predicted Input	Output	Predicted Mean	95% CI Low for Mean	95% CI High for Mean	Obtained Mean
**HMW** **CHITOSAN**	A(chitosan) = 0.51	Size (nm)	125.28	99.03	151.53	139.2
	B(TPP) = 0.25	PDI	0.228	0.200	0.250	0.246
		Zeta potential (mV)	30.1	26.7	33.5	26.8
**LMW** **CHITOSAN**	A(chitosan) = 0.20	Size (nm)	94.25	87.68	100.83	97.82
	B(TPP) = 0.41	PDI	0.256	0.220	0.29	0.223
		Zeta potential (mV)	16.1	15.1	17.1	15.9
**20 KDA** **CHITOSAN**	A(chitosan) = 0.80	Size (nm)	104.31	96.46	112.16	97.67
	B(TPP) = 0.34	PDI	0.299	0.250	0.350	0.271
		Zeta potential (mV)	21.9	20.6	23.2	21.8
**5 KDA** **CHITOSAN**	A(chitosan) = 0.56	Size (nm)	81.27	78.03	84.52	81.66
	B(TPP) = 0.41	PDI	0.257	0.240	0.280	0.245
		Zeta potential (mV)	20.4	19.6	21.2	19.8

## Data Availability

The data presented in this study are available on request from the corresponding author.

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
