# Peer review of "Design of Experiments to Achieve an Efficient Chitosan-Based DNA Vaccine Delivery System"

_pharmaceutics, 2021, doi:10.3390/pharmaceutics13091369_

Round 1

Reviewer 1 Report

The development of therapeutic vaccines to treat cancer is a rapidly growing field, thus studies evaluating possible DNA vaccines should always be promoted. In this work, authors applied the Design of Experiments in order to develop an efficient chitosan-based 2 DNA vaccine delivery system. Although, general the manuscript is well written, some comments should be answered before the publication.

Comment 1. Check for grammar and syntax errors

Comment 2. You should further discuss the results according to previously published manuscripts for chitosan nanoparticles

Comment 3. Stability studies to which protocol are based on? The stability of the nanoparticles is rapidly reduced by the time indicating low stability which is undesirable.

Comment 4. Physicochemical characterization is missing, FTIR, XRD, degradation as well as release etc should be performed in order to evaluate the chitosan properties

Comment 5.Could you please explain how the chitosan properties such as mucoadhesion or induced blood coagulation is important for vaccine delivery?

Author Response

The authors would like to acknowledge the careful revision and pertinent reviewers’ comments and the possibility to improve our manuscript, based on their constructive criticism. All the questions were answered in the attached document and the recommended modifications were made, being properly highlighted at yellow in the revised manuscript file. The authors look forward to hearing from you regarding the suitability of the revised manuscript for publication in the Pharmaceutics Journal.

Reviewer 2 Report

Authors used DoE tool to study the functional and effective plasmid DNA (pDNA) delivery system of four chitosan polymers with different MWs. However, many details need to be further revised. Based on this ground, it is recommended to major revision.

  1. How many times did the sizeand zeta-potential do?
  2. Table 1 lacks a more detailed comparative analysis and explanation. For example, why the sizes of LMW chitosan were smaller than those of
  3. The article has a lot of fitting results, is there any applicability, universality and application domain analysis? The authors are advised to add.
  4. Why is the stability test chosen 4℃?  Why not test the stability of body temperature ?

Author Response

(The authors gave the same response as above.)

Reviewer 3 Report

Chitosan can be useful as a nonviral vector for gene delivery. Although there are, many reports about chitosan-based  DNA vaccine delivery systems, the optimization and effect on transfection remain insufficient. The full understanding of the nanoformulation composition is very important, thus the lack of enough characterization of chitosan-based formulations may explain why the number of approved drugs with chitosan as an excipient is limited.

In this work, the authors show that the design of experiments proposed is a valuable tool that can be used to optimize nanoparticles formulation in a better way when compared with the common random experiment approaches.

Author Response

(The authors gave the same response as above.)

Round 2

Reviewer 2 Report

The author basically answered all the questions, which I agreed the publishing in this present form.